# 3D Dosimetry Based on LiMgPO_4_ OSL Silicone Foils: Facilitating the Verification of Eye-Ball Cancer Proton Radiotherapy

**DOI:** 10.3390/s21186015

**Published:** 2021-09-08

**Authors:** Michał Sądel, Jan Gajewski, Urszula Sowa, Jan Swakoń, Tomasz Kajdrowicz, Paweł Bilski, Mariusz Kłosowski, Anna Pędracka, Tomasz Horwacik

**Affiliations:** Institute of Nuclear Physics, Polish Academy of Sciences, 31-342 Krakow, Poland; jan.gajewski@ifj.edu.pl (J.G.); urszula.sowa@ifj.edu.pl (U.S.); jan.swakon@ifj.edu.pl (J.S.); Tomasz.Kajdrowicz@ifj.edu.pl (T.K.); Pawel.Bilski@ifj.edu.pl (P.B.); Mariusz.Klosowski@ifj.edu.pl (M.K.); Anna.Pedracka@ifj.edu.pl (A.P.); Tomasz.Horwacik@ifj.edu.pl (T.H.)

**Keywords:** 3D dosimetry, optically simulated luminescence (OSL), proton radiotherapy

## Abstract

A direct verification of the three-dimensional (3D) proton clinical treatment plan prepared for tumor in the eyeball, using the Eclipse Ocular Proton Planning system (by Varian Medical Systems), has been presented. To achieve this, a prototype of the innovative two-dimensional (2D) circular silicone foils, made of a polymer with the embedded optically stimulated luminescence (OSL) material in powder form (LiMgPO_4_), and a self-developed optical imaging system, consisting of an illuminating light source and a high-sensitive CCD camera has been applied. A specially designed lifelike eyeball phantom has been used, constructed from 40 flat sheet LMP-based silicone foils stacked and placed together behind a spherical phantom made by polystyrene, all to reflect the curvature of the real eyeball. Two-dimensional OSL signals were captured and further analyzed from each single silicone foil after irradiation using a dedicated patient collimator and a 58.8 MeV modulated proton beam. The reconstructed 3D proton depth dose distribution matches very well with the clinical treatment plan, allowing for the consideration of the new OSL system for further 3D dosimetry applications within the proton radiotherapy area.

## 1. Introduction

Modern radiotherapy (RT) techniques, such as intensity-modulated radiation therapy (IMRT), volumetric-modulated arc therapy (VMAT) or increasingly popular proton therapy, have enhanced the geometrical accuracy and complexity of the delivered dose distributions. Thus, spatial 2D- or even 3D-resolved measurements are required to properly validate complex treatment plans. These also imply the developments of new kinds of dosimetry techniques and materials. The current state-of-the-art for quality-assurance (QA) systems in modern RT are the ionization chambers. On the other hand, passive methods such as those based on optically stimulated luminescent (OSL) detectors, in the form of small chips, are widely applied, e.g., for in vivo dosimetry (IVD), personal dosimetry or phantom measurements. However, these will always give only one point (1D) of information. For instance, OSL chips are increasingly utilized for the IVD of complex radiation therapy techniques such as IMRT, VMAT or ion beam therapy [1,2,3]. However, as the clinical reports show, patient entrance doses obtained for the IVD of complex RT techniques vary significantly due to the chip’s placement errors. As a result, the dosimetric uncertainty can be even doubled [4]. Therefore, one of the solutions to avoid dose discrepancies is to migrate from point-based in vivo dosimetry to planar or volumetric methods, enabling spatial dose measurements. The interest in developing high precision 2D/3D dosimeters has been more and more present, especially in the past few years, mostly due to the applications of modern CCD/CMOS cameras for capturing high resolution 2D images. Concerning the 2D OSL dosimetry for RT and diagnostic imaging applications, a short list of different OSL materials that have been applied can be found as follows: SrS [5], BeO [6], KCL [7] or Al_2_O_3_:C [8,9].

Recently another new and promising approach to the 2D OSL dosimetry system has been presented. By using a self-developed optical imaging setup consisting of an illuminating light source (using blue LED’s) and a highly sensitive CCD camera, together with a prototype of the 2D flat sheet foils created by embedding the OSL material in powder form into a transparent silicone elastomer matrix, the 2D/3D proton depth dose distributions have been verified [10]. Two kinds of OSL active material have been tested so far. In the first attempt, optimized at an angle of its OSL properties, the LiF:Mg,Cu,P (MCP) powder was embedded into the silicone host [10,11,12,13]. Finally, another promising OSL material, namely the LiMgPO_4_ (LMP), has been used as OSL powder embedded into a silicone host. Its promising dosimetric properties, a high radio-sensitivity, broad linear dose response (up to 1 kGy), good repeatability and acceptable fading [14,15,16,17,18,19], allow for considering the LMP material as the future alternative for the commercially available OSL materials. Recently, by using a 2D LMP-based silicone host and a self-developed optical imaging setup for 2D OSL readouts, a real 3D depth dose distribution of the single, unmodulated proton Bragg peak has been measured and compared within a standard proton QA procedure using the ionization chamber [18].

The present work shows a step forward into the future clinical applications of the recently developed OSL technology and is focused on investigating the possibility of using the LMP-based flat sheet silicone foils and self-developed optical readout system to verify the real 3D clinical proton treatment plan, prepared for eyeball tumors using the Eclipse Ocular Proton Planning system (by Varian Medical Systems). Direct measurements of the proton depth dose distribution, using a dedicated patient irregular collimator and 58.8 MeV modulated proton beam, were possible using specially constructed lifelike eyeball phantoms, constructed from 40 flat sheet circular LMP-based silicone foils, stacked together inside the PMMA (Poly(methyl methacrylate)) holders and placed directly behind a spherical phantom made by polystyrene, all to best simulate the dimensions of a real human eyeball. The ability to register a delivered spatial 3D proton radiation dose distribution in a quick and easy readout procedure (based on OSL phenomena) demonstrates the true potential of the 2D OSL technology, as well as enables its future development in modern radiotherapeutic techniques such as proton radiotherapy.

## 2. Materials and Methods

### 2.1. D OSL Silicone Foils

The reported investigations were realized using the 2D flat sheet silicone foils of a size of approx. 20 (±0.1) mm in diameter and 0.54 (±0.3) mm thickness, made by mixing the self-synthesized LiMgPO_4_ (LMP) powder into a commercially available silicone elastomer matrix—the SYLGARD^®^ 184 Silicone Elastomer Kit from Dow Corning [20]. The preparation procedure comprises mixing homogeneously the LMP powder into a silicone elastomer matrix (at 1:2 weight ratio), pouring it into the special aluminum holders and placing it into a vacuum desiccator in order to remove any pre-existing air bubbles (see Figure 1). The LMP OSL material doped with Tb (1.2 mol%) and B (10 mol%) was synthesized in powder form by a solid-state reaction in air (grains of a size of approximately 60 µm) [16]. The detailed description of the sample preparation can also be found in our previous publication [18].

### 2.2. Proton Treatment Plan Using the Clinical Treatment Planning System (TPS)

A proton treatment plan was prepared using the Eclipse Ocular Proton Planning (EOPP) program, version 13.5.01, by Varian Medical Systems. A sample tumor shape was drawn inside a spherical eye model. The tumor parameters were as follows: location—left eye, backward from the equator, in the back pole of the eyeball; distance to the disc edge −0.80 mm; distance to the macula edge −0.85 mm; cover up the macula, maximum tumor base diameter −8.50 mm; tumor height from retina −2 mm. In order to follow the eyeball phantom capabilities, the therapeutic eye position was set up to be looking straight ahead (polar gazing angle of 0 deg and azimuthal gazing angle of 0 deg). The resulting beam parameters were 29 mm of range penetration and 10 mm of range modulation. The shape of the collimator aperture was automatically defined to surround the target area with a defined margin of 2.5 mm. The treatment field was aligned with the tumor center (Figure 2). The eye center was shifted relative to the isocenter: x = 1.02 mm, y = 1.83 mm, z = 10.50 mm (shown in Figure 2 and Figure 3). Figure 4 shows the brass collimator used for proton irradiations, together with its contour dimensions.

### 2.3. Eye-Ball Phantom Construction for TPS Verification

Figure 5 shows the eyeball phantom used for proton irradiation. The phantom was constructed by using a spherical cap made of polystyrene, of a maximum thickness of 10 mm and a diameter of 11.5 mm. Forty 2D LMP silicone foils were stacked together and placed tightly inside a cylindrical hole created by tightly stacking 10 of the PMMA holders shown in Figure 1. Thus, the total thickness of the phantom comprised 10 mm thickness of the polystyrene hemisphere and 40 times 0.54 mm, i.e., the average thickness of the LMP foil.

In order to check the water-equivalent thickness (WET) of the LMP foil material, as well as for the spherical phantom, a separate experiment has been performed. The WET parameter relates to the thickness of the water layer, expressed in g/cm^2^, which causes the same loss of proton energy as in a given material with a given thickness. The obtained WET shows that 1 mm of the LMP foil as well as 1 mm of a hemispherical phantom, causes the same energy loss as 1.05 mm of water. This was also exactly the same value of human eyeball WET used when preparing a treatment plan using the clinical TPS. Finally, taking into consideration the total thickness of the phantom and the WET of the polystyrene hemisphere and LMP foils, the total available range for proton measurements of the experimental phantom expressed in water millimeters is fixed to 33.31 mm (31.72 × 1.05).

Figure 6 shows the experimental setup, including the phantom mounted on a therapeutic chair in the treatment room and the exit of the proton snout. The eye-ball phantom with the LMP foils was irradiated with a dose of 60 Gy with a field formed by an individual brass collimator designed in accordance with the therapy plan.

### 2.4. Proton Beam Dosimetry

Irradiation of the eyeball phantom was performed at the Proton Eye Radiotherapy Facility at the IFJ PAN with a 58.8 MeV modulated and collimated proton beam, available from the AIC144 isochronous cyclotron [21]. The dosimetry has been carried out by a plane-parallel ionization chamber in a water phantom according to the IAEA TRS-398 protocol [22]. A PTW Markus ionization chamber model TM23343 connected to the reference class electrometer PTW UNIDOS was used as reference dosimetry to determine the dose in water and to collect depth-dose distribution along the central axis of the modulated proton beam for the parameters specified in the treatment plan. In order to verify the consistency of the irradiation conditions with the irradiation plan, the depth-dose distribution has been measured with the Markus ionization chamber connected to the electrometer. The measurement was performed in a water phantom in the collimated beam with a diameter of 25 mm and with the modulation of 10 mm (No. 010). Figure 7 shows the relative depth-dose distribution measured in water for a range modulator with nominal modulation and range of 10 mm and 28.96 mm, respectively.

### 2.5. Image Readout System

The measurements of the 2D OSL signals have been performed using a self-developed system, consisting of illuminating light source (the CoolLED blue LEDs 470 mm) and optical detection system (the Apogee Alta U47–UV CCD camera), supported with a filter set for discrimination between excitation (a band-pass filter 470/40 nm) and emission light (4 mm of the U-340 filter along with interference short-pass 468 nm filter). Detailed information about the system configuration set can be found in our previous publication [14]. Figure 8 presents the result of spectrally resolved measurement, together with a comparison between the used CCD camera quantum efficiency, the LED emission spectrum and the applied filters’ transmission windows. A separate experiment was performed to check the LMP OSL emission spectrum. Specially prepared smaller LMP sample foils (6 mm in diameter), irradiated with ^90^Sr/^90^Y beta source to a dose of approx. 5 kGy have been measured using the Ocean Optics QE pro 00689 spectrometer in conjunction with the blue LEDs (470 nm) for stimulation and the same filter set as for the spatially resolved measurements. The measured LMP OSL emission spectrum extends from 330 nm to 400 nm, with two peaks at 355 nm and 380 nm (see Figure 8).

### 2.6. Image Data Acquisition

Two-dimensional OSL images were captured from each single irradiated LMP foil with a CCD camera during acquisition time of 30 s using an optical system described above. Images were captured using the Fiji open-source software [23]. Next, in the context of signal processing, images were corrected and analyzed in the Python environment, importing, among others, the functionalities FRED tools module [24,25,26]. Pixel intensities depend on several parameters and may be reflected in the camera setup parameters, image distortions or powder distribution inside the silicone matrix. Each measured image was corrected in the following procedure, presented in detail in Figure 9. First, the background, i.e., an image of non-irradiated sample foil, was subtracted. Then, the image was clipped to the area of the detector, found on a live view image of the detector (Figure 9—panel a), i.e., an image acquired in 0.6 s in normal white light. The original image was then corrected using a flat-field image, i.e., an image acquired with 30 s, using blue LEDs light (Figure 9—panel b). The inhomogeneity of the detector response was minimized using an individual reference image (IRI) of the foil, irradiated previously with uniform dose distribution in the whole area of the sample using e.g., ^60^Co gamma source gamma—Theratron 780E (Figure 9—panel c and d). After readout, the IRI is corrected in the same way as the measurement image and is used for correction of each pixel value. Finally, the corrected images are using for comparison with the clinical data obtained from the ionization chamber.

## 3. Results

### Verification of the TPS Using the 2D LMP Foils

Figure 10 (panel a) shows the relative proton dose distribution rescaled to water, calculated by the TPS (see also Figure 3), presented together with the position of the spherical phantom used during irradiation. In panel b, the comparison of the proton depth-dose distribution (the so-called spread-out Bragg Peak, SOBP) measured with the Markus ionization chamber, extracted from the treatment plan through the central point and indicated by a red line in Figure 10 (panel a) have been shown.

Figure 11 shows reconstructed 3D proton depth-dose distributions, obtained from the stack of forty LMP foils placed behind the spherical phantom (panel a) and for the TPS data (panel b). Each image has been rescaled to the maximum value in the 3D OSL dose distribution. Additionally, a comparison of the longitudinal line profiles, plotted through the image centre (black solid line visible on both 3D images) were presented (Figure 11, panel c). The signal obtained from the LMP foils (blue points) increases in the plateau region, then decreases in the SOBP region, showing an under-response, also known as a quenching effect or lower luminescent efficiency. This is similar to what has been observed previously for the other OSL materials, and it is related to the ionization density of the heavy charged particles that pass through the irradiated material [27]. In turn, the observed foils’ over-response in the proximal region of the SOBP is the result of the applied normalization of the measurement to the maximum value of the whole 3D OSL signal distribution. Nevertheless, the obtained proton dose-response of the LMP foils suggests the need to further investigate the relative luminescent efficiency of the LMP material for protons in a separate experiment, especially that to our knowledge, such data are not available at present. A video showing the 2D OSL images from each of the 40 LMP foils placed behind the spherical phantom captured after proton irradiation, compared with the same images extracted from the TPS, and a comparison of the longitudinal cross-sections through the centre of the whole 3D OSL and TPS image has been shown in the Appendix A.

Figure 12 shows the comparison of lateral cross-sections (2D OSL images), extracted from the 3D relative dose images, obtained from the LMP foils (Figure 11 panel a) and from the TPS (Figure 11 panel b) for three depths: 24 mm (middle of the SOBP), 29 mm (range of the SOBP at 90% of the distal fall-off) and 29.8 mm (range of the SOBP at 10% of the distal fall-off), indicated with black dashed lines in Figure 11 (panel a and b). Additionally, in panel c, comparisons of the line profiles along the *X*-axis for the corresponding depths have been shown.

## 4. Discussion

The reconstructed 3D proton dose distribution obtained from a stack of 40 silicone 2D OSL foils (presented in Figure 11) is a first and very promising step, producing the possibility to verify a delivered real 3D proton treatment plan, prepared with the clinical TPS software. It should be emphasized that both the self-developed optical detection systems for retrieving the 2D OSL signals, as well as the LMP silicone foils, are just a working prototype system that has been constructed from the readily available parts for the purpose of demonstrating, in practice, the anticipated signal levels and thus has a character of a proof-of-concept study. Firstly, the technology gives the ability to reproduce the shape of a spatial proton dose distribution. Secondly, the lateral cross-sections, extracted from each single 2D OSL signal captured from the LMP foil at different depths (Figure 12), and compared with the same profiles extracted from the TPS, gives a very good agreement. It should also be noted that the profiles presented in Figure 12 have been normalized to the maximum value of the whole 3D OSL signal distribution. This was done to mimic the normalization of the 3D dose distribution calculated by the TPS. In order to investigate the applicability of the 2D LMP foils for the field lateral shape measurements, the signal from the foils can be normalized differently, i.e., to the maximum value of a cross-section at given depth. Such comparison was shown in Figure 13, where both the 2D OSL and the TPS lateral cross-sections had been normalized to the maximum value. The obtained comparison of the lateral cross-sections and resulting isodoses presented in panels c and d of Figure 13 allow for considering the 2D LMP foils as a promising tool for beam shape measurements in small radiation fields used in eyeball cancer proton radiotherapy.

Nevertheless, in order to exploit the true 3D potential of the OSL technology, e.g., during the QA tests before proton treatment, further optimization should be performed. The dependence of the LMP OSL luminescence response on proton energy, as mentioned in the Section 3 and presented in Figure 11 (panel c), is not linear and showed an under-response, suggesting the need to apply efficiency corrections depending on proton energy, especially since, to our knowledge, such data are not available at present for LMP material. These measurements should be done with monoenergetic uniform proton fields for various energies constituting the SOBP field. This is the plan for the next steps of the study. However, for the purpose of this work and based on our previous experiment [18], performed using the same type of the LMP foils and irradiated with unmodulated 58.8 MeV proton beam, a simple efficiency correction has been applied. Figure 14 (panel a) shows the relative dose response, measured with the Markus ionization chamber in water (blue solid line) and directly from the thirteenth 2D LMP foils (red points) placed at various depths and irradiated with a single 58.8 MeV proton beam (data according to the [18]). Based on these results, the depth dose calibration factor has been calculated as the ratio of the relative dose measured by the Markus ionization chamber (blue solid line) and the 2D LMP foils (red points). Figure 14, panel d, shows the obtained calibration factors, together with the calibration curve, fitted in order to determine the calibration factors in the whole SOBP range. Finally, the corrected results are present in Figure 14 panel c for the depth line profiles through the centre of the corrected (green points), for the uncorrected (red points) 3D images and for the lateral profiles at depth 24 mm on the panel f.

## 5. Conclusions

The new and promising 2D OSL dosimetry system, based on the prototype LMP silicone foils, and the self-developed optical imaging system have been used to verify a real proton treatment plan prepared for eyeball cancer by using a dedicated Eclipse Ocular Proton Planning system (by Varian Medical Systems). The direct verification of the 3D proton dose distribution was performed using a specially designed eyeball phantom, constructed from 40 LMP silicone foils stacked and placed together behind a spherical cap made of polystyrene and reflecting the curvature of the real human eyeball. Such a constructed eyeball phantom has been irradiated with 58.8 MeV modulated and collimated proton beam. The obtained results show a good agreement of the reconstructed 3D OSL proton depth dose distribution with the clinical treatment plan. The lateral cross-sections extracted from each single LMP foil gives useful information about the shape of the used proton field, allowing for consideration of the presented 2D OSL system for beam-shaped measurements in small radiation fields, such as those used in eyeball cancer proton radiotherapy. However, further investigations are necessary within the LMP’s relative luminescence response to protons and its dependence on proton energy in order to enhance the true application character of the technology and for the quality assurance of the cancer treatment in proton radiotherapy. Nevertheless, it should be emphasized that the ability to register a delivered spatial 3D proton radiation dose distribution in a quick and easy readout procedure (based on OSL phenomena) makes the new system one of a state-of-the-art tool in the future 3D dosimetry applications.

## Figures and Tables

**Figure 1 sensors-21-06015-f001:**
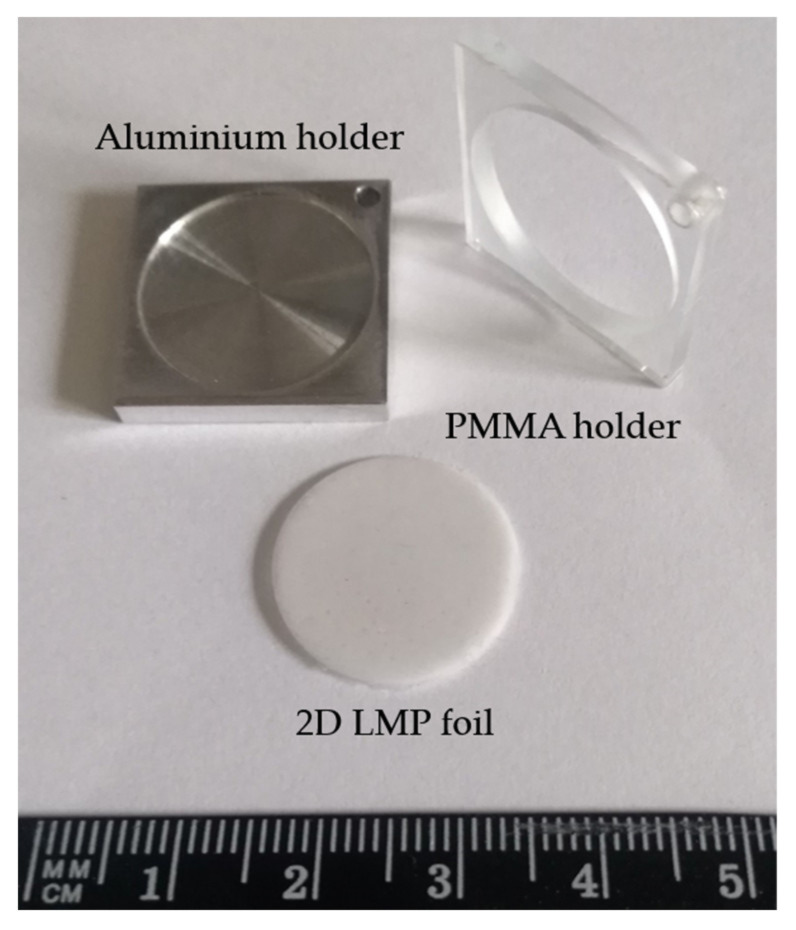
The prototypes of the 2D LMP flat sheet silicone foil of a size of 20 (±0.1) mm in diameter and 0.54 (±0.3) mm thickness produced at the special aluminum holder (upper left corner) by homogeneously mixing the self-synthesized LMP powder into the SYLGARD^®^ 184 silicone elastomer matrix at a 1:2 weight ratio. Additionally, on the right-hand side, 1 of the 10 PMMA holders with a hole of a size of LMP foil, used for proton irradiation, is shown.

**Figure 2 sensors-21-06015-f002:**
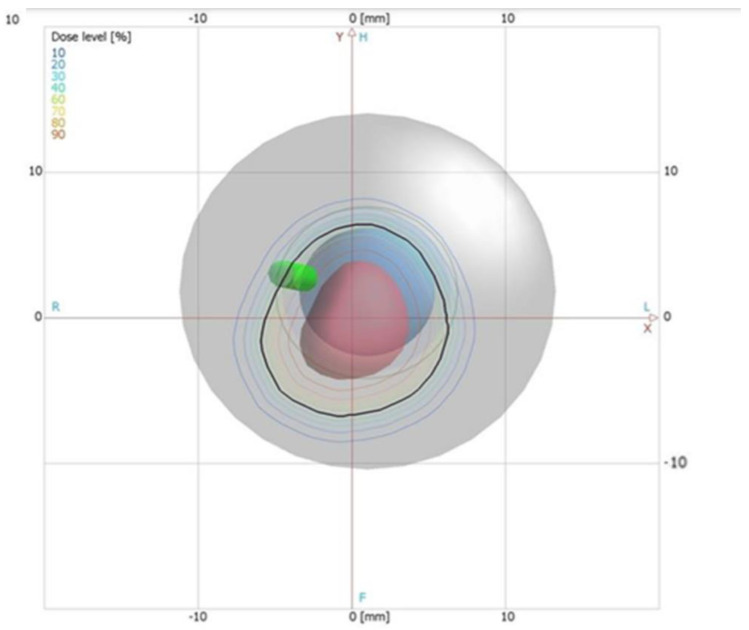
Beam’s eye view, eye front orientation from the Eclipse Ocular Proton Planning system. The black line is the shape of the collimator aperture (see also Figure 4) surrounding the tumor shape (red area). The isodoses represent the dose distribution in the eye on an X-Y section through isocenter.

**Figure 3 sensors-21-06015-f003:**
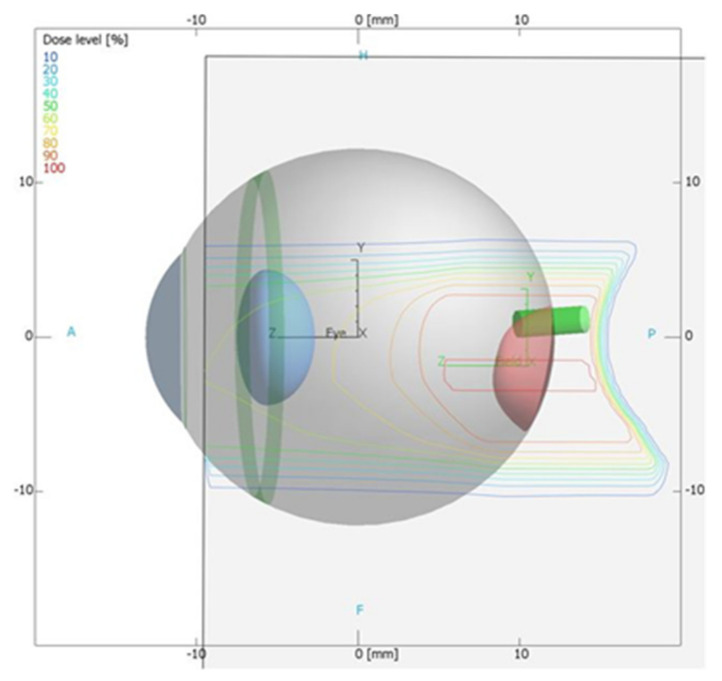
Lateral view, lateral orientation from the Eclipse Ocular Proton Planning system. The eye center (X-, Y- and Z-coordinates of the center point of the eye) is in black, the isocenter (X-, Y- and Z-coordinates) is in green. The isodoses represent the dose distribution on a Y-Z section though the isocenter.

**Figure 4 sensors-21-06015-f004:**
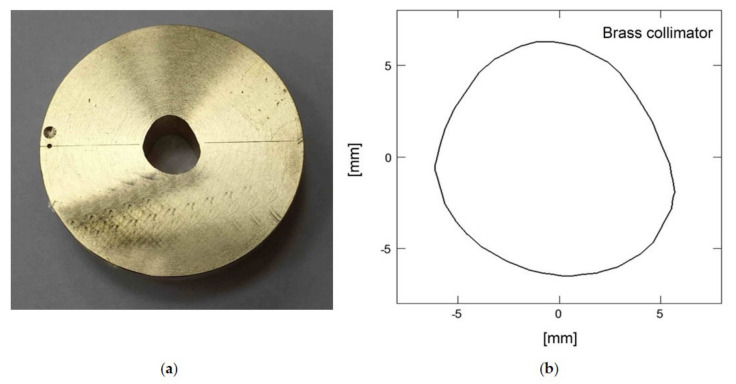
(**a**) Picture of the brass collimator used for proton irradiations; (**b**) The contour of the brass collimator with dimensions.

**Figure 5 sensors-21-06015-f005:**
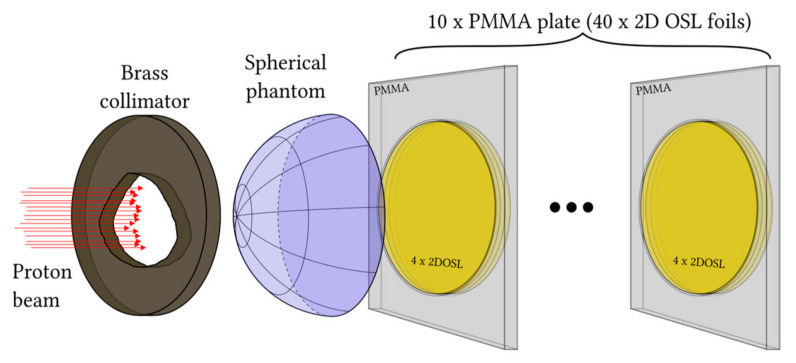
The construction of the eyeball phantom used for proton irradiations. The 40 silicone 2D LMP foils stacked tightly together, inside the hole constructed from the 10 PMMA holders behind a spherical phantom.

**Figure 6 sensors-21-06015-f006:**
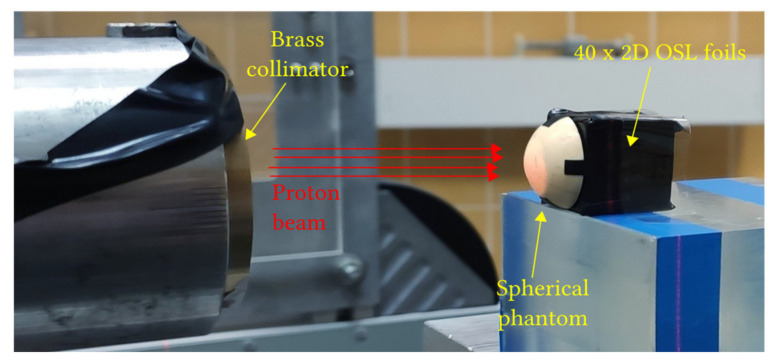
Experimental setup: 40 silicone 2D LMP foils attached to 10 PMMA holders behind a spherical phantom, mounted on the top of the therapeutic chair in the treatment room, irradiated with 58.8 MeV modulated and collimated proton beam.

**Figure 7 sensors-21-06015-f007:**
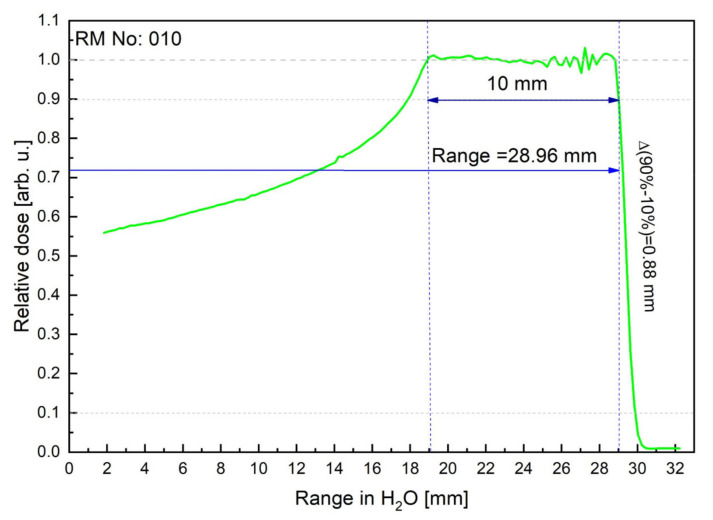
Proton depth dose distribution (the so-called spread-out Bragg Peak, SOBP) for the Markus ionization chamber (green solid line), measured using a 3D scanner and 10 mm collimator. The measured parameters: modulation = 10 mm; Range, R_90_ = 28.96 mm; Distal fall-off Δ(R_90%_ − R_10%_) = 0.88 mm.

**Figure 8 sensors-21-06015-f008:**
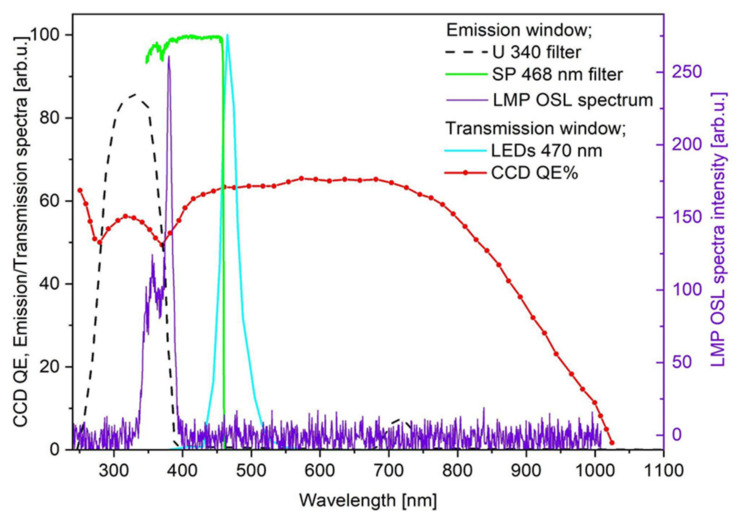
Comparison between the CCD camera quantum efficiency, the LEDs emission spectra (470 nm) and the 468/SP + U340 filters transmissions (all data left hand scale). The measured OSL emission spectra of the LMP material (irradiated with ^90^Sr/^90^Y beta particles), extends from 330 nm to 400 nm, with a maximum two peaks at 355 nm and 380 nm (right hand scale).

**Figure 9 sensors-21-06015-f009:**
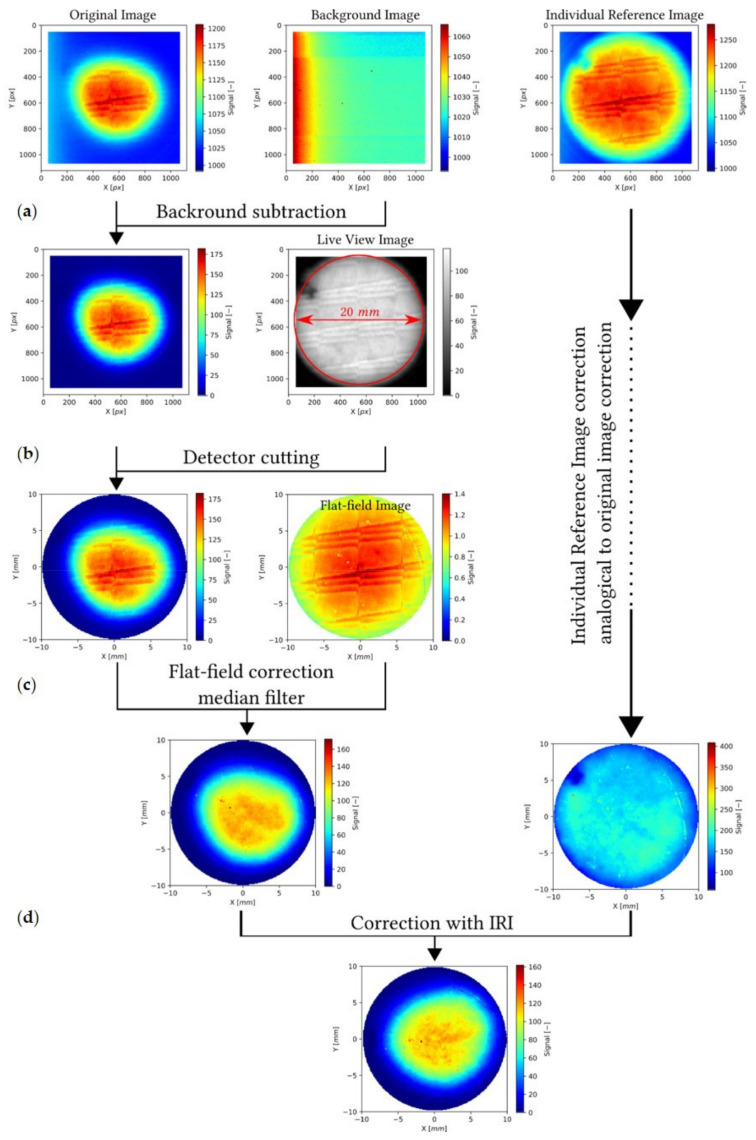
(**a**) Image correction procedure, consisting of background subtraction; (**b**) Cutting based on the circular foil found in a live view image; (**c**) flat-field correction and median filtering. The image of individual pixel response, i.e., readout of the same foil after exposure to uniform radiation field is corrected in the same procedure; (**d**) Afterwards, the corrected measurement image is corrected with the individual reference image.

**Figure 10 sensors-21-06015-f010:**
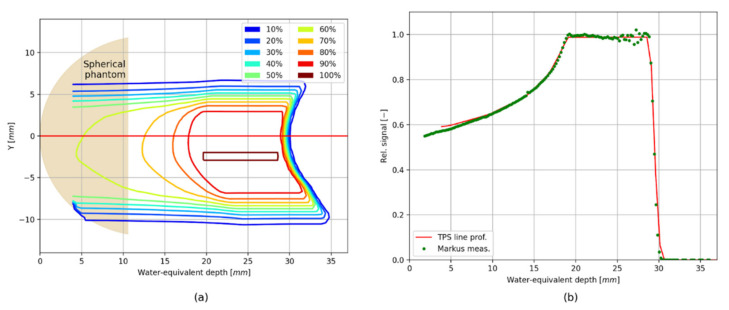
(**a**) Isodoses of the longitudinal cross-section extracted from 3D relative dose image calculated by the TPS 3D (**a**) rescaled to water; (**b**) Longitudinal line profile of the treatment plan through the central point (indicated by a red line) with measurements made by a Markus chamber in water (green points—panel a).

**Figure 11 sensors-21-06015-f011:**
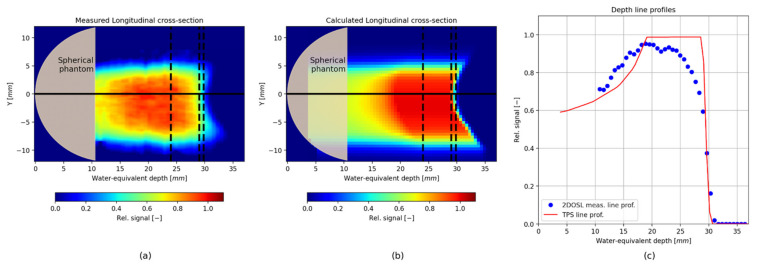
(**a**) The reconstructed from the 3D dose distributions longitudinal cross-section through the image centre extracted from the stack of forty 2D LMP foils, (**b**) 3D relative dose image calculated by the TPS, (**c**) A comparison of longitudinal line profiles plotted through the image centre (black solid lines visible on both 3D images).

**Figure 12 sensors-21-06015-f012:**
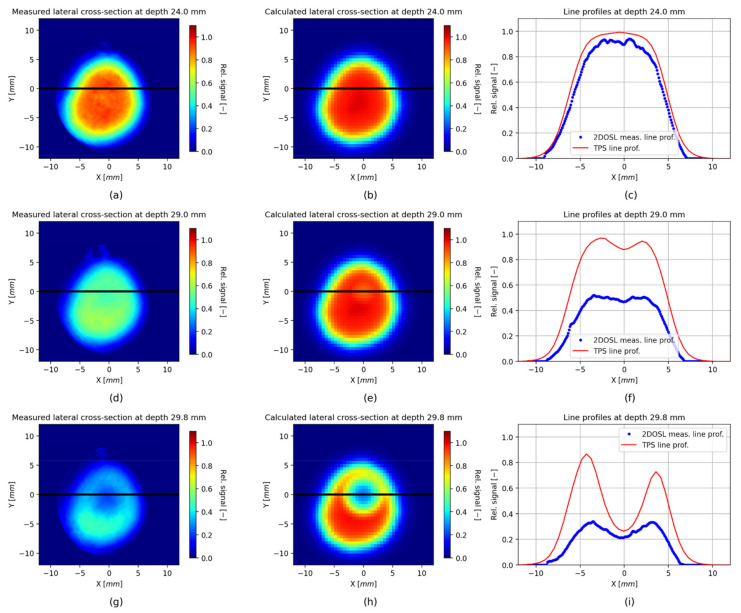
Lateral cross-sections extracted from the 3D relative dose image measured with the stack of 2D LMP foils (**a**,**d**,**g**) and from 3D relative dose image calculated by the TPS (**b**,**e**,**h**) at depths of 24 mm (middle of the SOBP), 29 mm (range of the SOBP at 90% of the distal fall-off) and 29.8 mm (10% of the distal fall-off), respectively; (**c**,**f**,**i**) Comparisons of line profiles along the *X* axis for corresponding depths. The depths are indicated with black dashed lines in Figure 11 (panel a and b).

**Figure 13 sensors-21-06015-f013:**
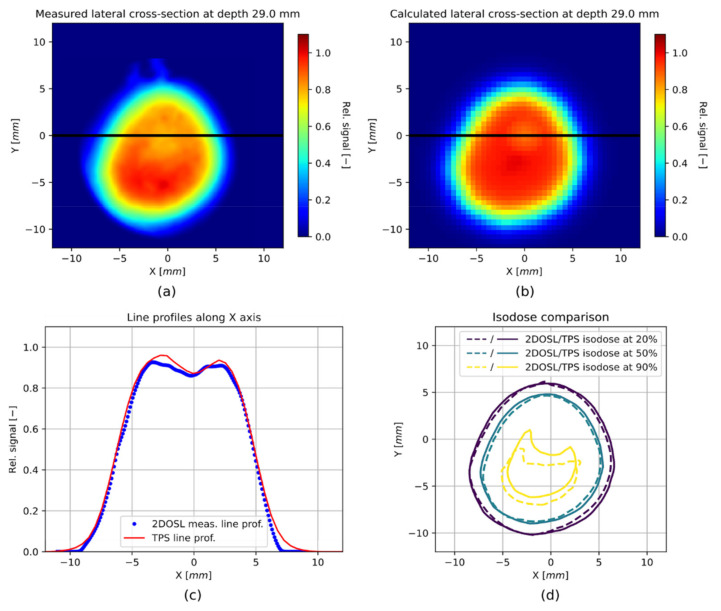
(**a**) Lateral cross-sections extracted from the 3D relative dose image measured with the stack of 2D LMP foils; (**b**) Lateral cross-sections from 3D relative dose image calculated by the TPS at depth 29 mm (range of the SOBP at 90% of the distal fall-off), both normalized to the maximum value of the cross-section; (**c**) A comparison of line profiles along the *X* axis; (**d**) A comparison of isodoses at 20%, 50% and 90% of the maximum value of the cross-section.

**Figure 14 sensors-21-06015-f014:**
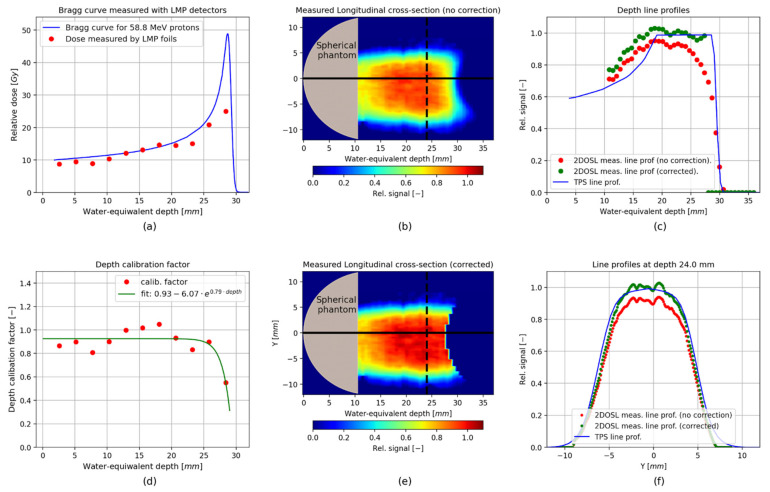
(**a**) Depth dose correction using results of dose measurements with 2D LMP foils of a monoenergetic proton beam, measured with a Markus ionization chamber, along with the corresponding measurements with 2D LMP foils (data according to the [18]); (**b**) Uncorrected longitudinal cross-section through the image centre; (**c**) Comparison of longitudinal line profiles through the image centre; (**d**) Depth correction factor calculated as the ratio of the relative dose measured by the Markus chamber and the 2D LMP foils; (**e**) Longitudinal cross-section through the centre of the corrected 3D image (**f**) Line profiles along *Y* axis for 3D dose calculated in TPS and 3D uncorrected and corrected images measured with the stack of 2D LMP foils.

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
