# Peer review of "3D Dosimetry Based on LiMgPO4 OSL Silicone Foils: Facilitating the Verification of Eye-Ball Cancer Proton Radiotherapy"

_sensors, 2021, doi:10.3390/s21186015_

Round 1

Reviewer 1 Report

  1. In the introduction it would be useful to summarize LMP based flat sheet silicone foils basic dosimetric parameters, of repeatability, dose-response, dose rate dependance and fading from reference 14 or other applicable references.
  2. Please clarify the reference to chapter 3 and 4 on pp. 8, 9,13.
  3. In Fig. 11 why weren’t additional longitudinal cross section taken that represent more of the image such as at 15 cm and  20 cm?  What is the explanation of the depth dose shift in 3D relative dose image measured with the stack of 2D LMP foils (a) and from 3D relative dose image calculated by the TPS 3D ?  Typically in 2D/3D dosimetry evaluations the TG119 gamma criteria have been used in these comparisons.
  4. In Figure 14a. The authors acknowledge that there are additional studies needed for correcting for the Bragg peak response and that they used a “a simple efficiency correction has been applied”. There are no additional details on that correction.
  5. This is a work in progress

Reviewer 2 Report

This study is meaningful for improvement of the OSL technique with respect to 2D and 3D dosimetry and can largely contribute to the in vivo dosimetry in radiotherapy. Experimental 3D dosimetry of small radiation fields is still a challenge, especially in the case of proton beam radiotherapy. Therefore, the presentation of these new results on this issue will be of considerable importance.

The authors can optionally address the following comments:

1) On page 3, figure 4 is cited before figures 2 and 3.

2) Apparently, there are some commas misplaced in the text, such as:

  • The interest in developing high precision 2D/3D dosimeters, has been more…
  • … the observed foils over-response in the proximal region of the SOBP, is the result of…
  • It should be also noted that the profiles presented in Figure 12, have been normalized…
  • …with the self-developed optical imaging system, has been used…
  • The lateral cross-sections extracted from each single LMP foil, gives useful…

3) The following sentence seems to need improvement.

The comparison of the line profiles and isodoses, allowing to consider the 2D LMP foils for the beam shape measurements in small radiation fields used in eye-ball cancer proton radiotherapy.

4) The authors refer to “chapter 3 and 4”. I believe it would be more appropriate to replace "chapter" with "section".

I think the work can be published by Sensors.

Round 2

Reviewer 1 Report

Author has adequately modified h]the manuscript